# Laboratory Study of Turbulent Mass Exchange in a Stratified Fluid

Andrey G. Zatsepin *, Valerii V. Gerasimov * and Alexander G. Ostrovskii

Shirshov Institute of Oceanology, Russian Academy of Sciences, Nakhimovskiy Prospect 36, 117997 Moscow, Russia; osasha@ocean.ru
* Correspondence: zatsepin@ocean.ru (A.G.Z.); gerasimov.vv@ocean.ru (V.V.G.)

**Abstract:** In this study, a laboratory experiment was conducted to investigate quantitatively turbulent exchange between two quasi-homogeneous layers of equal thickness and different density (salinity), as well as the fine structure of the density transition zone (interface) between the layers. The fluid was continuously stirred by a system of horizontally oscillating vertical rods, piercing through both layers and producing vertically homogeneous turbulent impact in a two-layered fluid. In every experimental run, the stirring process was carried out continuously from certain initial state up to the complete mixing of the layers. The buoyancy flux between the layers was estimated using the data on time changes of the salinity in both upper and lower layers. The fine structure of density interface was measured by vertically profiling conductivity microprobe. The results were presented in a dimensionless form and analyzed depending on two dimensionless parameters as follows: the Richardson number, $Ri$, and Reynolds number, $Re$. It was found that if $Ri > Ri^*(Re)$ where $Ri^*$ is the critical Richardson number, the interface exists in "sharpening" mode and in "eroding" (diffusive) mode if $Ri < Ri^*(Re)$. The maximum mixing efficiency was achieved at critical Richardson number, when the density interface was in a transition state between the sharpening and diffusive modes.

**Keywords:** laboratory experiment; two-layered fluid; turbulent stirring; mixing efficiency; interface thickness





## 1. Introduction

It is well known that the ocean stratification is almost never "smooth" and is characterized by the fine structure (FS), which is manifested as alternation of layers and interlayers with different values of vertical gradient of temperature, salinity, density, etc. According to Konstantin N. Fedorov, the well-known researcher of the FS in the ocean, the FS can be treated as "signature" of the mixing processes, both vertical and horizontal [1,2]. It should be noted that sometimes such signature is "calligraphic" when a regular FS is formed, consisting of a system of quasi-uniform layers separated by sharp density interfaces. Examples of such structures can be found in the above-mentioned books [1,2].

From general considerations, it follows that the type of mixing (convection, double-diffusive convection, velocity shear and shear-free turbulence, etc.) should not have a decisive effect on the final result—the transformation of an initially smooth vertical density gradient into a step-like structure profile. Rather the theoretical and laboratory modeling data show that vertically homogeneous turbulent mixing under certain conditions transforms a continuous density stratification into a step-like profile. The basic process of thin layering of the stratified fluid due to turbulent stirring was proposed by Phillips and Posmentier [3,4]. In a strongly stratified fluid, the process involves instability of vertical turbulent exchange. It was shown that at the Richardson number greater than some critical value, any local increase in the vertical density gradient tends to grow further on. This is due to the weakening of the vertical turbulent mass flux in the region of the increased gradient. Below and above this layer the gradient decreases while the mass flux increases. Over time, in the areas of turbulent mass flux decrease, sharp density interfaces are formed

(similar to that how traffic jams occur in places of narrowing of the road). They suppress turbulence, and in the vicinity of them, in the areas of the turbulent mass flux increase, quasi-uniform layers are formed.

The possibility that the initially linearly density-stratified fluid exposed to vertically homogeneous stirring of undergoes the FS layering was demonstrated in laboratory experiments [5–8] (see Figure 1) as well as by means of mathematical and numerical modeling [9–11].

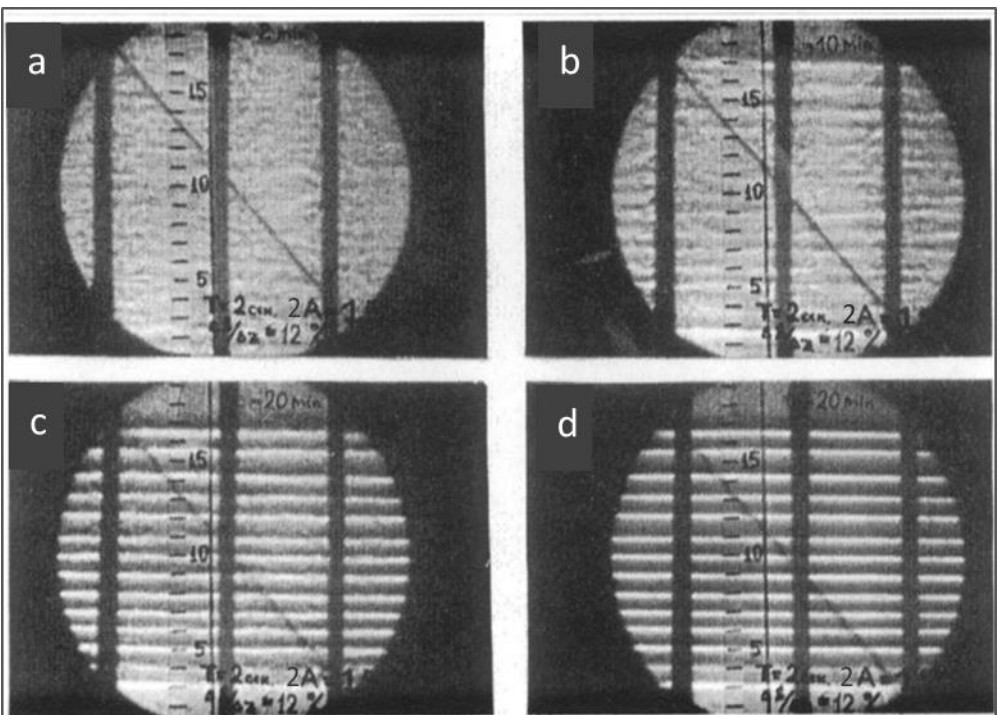

**Figure 1.** Consecutive shadowgraphs illustrate the formation of the step-like fine structure during long-term stirring of linearly stratified water column by horizontally oscillating vertical rods. The shadowgraphs were obtained during the experiment as follows: (**a**) top left at 10 min, (**b**) top right at 30 min, (**c**) bottom left at 240 min, (**d**) bottom right at 630 min. Experiment setup parameters were as follows: $A = 0.75$ cm, $d = 0.65$ cm, $T = 2$ s, $N^2 = (d\rho/dz)g/\rho = 8.4 \text{ s}^{-2}$, where $A$ is amplitude, $d$ is diameter of the rod, $T$ is period of rod oscillation, $N$ is Brent-Väisälä frequency, $d\rho/dz$ is density gradient, $\rho$ is water density, $g$ is gravity acceleration).

In this study, which can be considered as a development of earlier work [8], the method of physical (laboratory) modeling is used to investigate quantitatively: (i) the turbulent exchange between two quasi-homogeneous layers of different salinity having equal thickness; and (ii) the structure of the density transition zone between the layers. The layers are continuously stirred by a fixed system of horizontally oscillating vertical rods piercing through both layers. The movement of the rods produces vertically homogeneous turbulent stirring of a two-layered fluid media. The experiment is different from the experiment of Linden [12] and Krylov and Zatsepin [13] which were focused on turbulent mass transfer through the density interface between the layers forced by the freely falling grid or vertically oscillating grids with square rods.

The objective for this work was to study the relationship between the structure of the water density boundary layer between two layers of different salinity and the vertical buoyancy flux induced by continuous and vertically uniform turbulent impact. The experiments were carried out in a wide range of the values of Richardson number in order to demonstrate abrupt transition from the mode of maintaining the density interface in a sharpened state to the mode of its erosion—diffusion expansion. The relevance of this transition to the Phillips-Posmentier model is discussed.

## 2. Materials and Methods

The experiments were carried out on a laboratory setup, the schematic diagram of which is shown in Figure 2. The laboratory tank made of organic glass was of an internal size of $36 \times 15.5 \times 25$ cm$^3$. Horizontally oscillating metallic rod above the tank was oriented along the longer side of the tank. Six grids were fixed evenly at this rod with a $L = 5.5$ cm distance between the adjacent grids. Each grid consisted of six vertical glass rods with the diameter $d = 0.65$ cm with a distance $M = 2.8$ cm between each other (the mesh size). The glass rods were submerged in the water down to the 0.5 cm thick bottom layer of the tank. The metallic rod, on which the grids with the glass rods were fixed, was connected to a DC electric motor with an eccentric, which provided horizontal oscillation of the rod and the grids. By changing the supply voltage to the motor, the oscillation period $T$ of the grids could be changed in the range from 1.3 to 6.7 s. By changing the attachment point of the metallic rod to the eccentric it was possible to vary the amplitude A of oscillations in the range from 0.5 to 1.8 cm. The measurement data were collected in a personal computer.

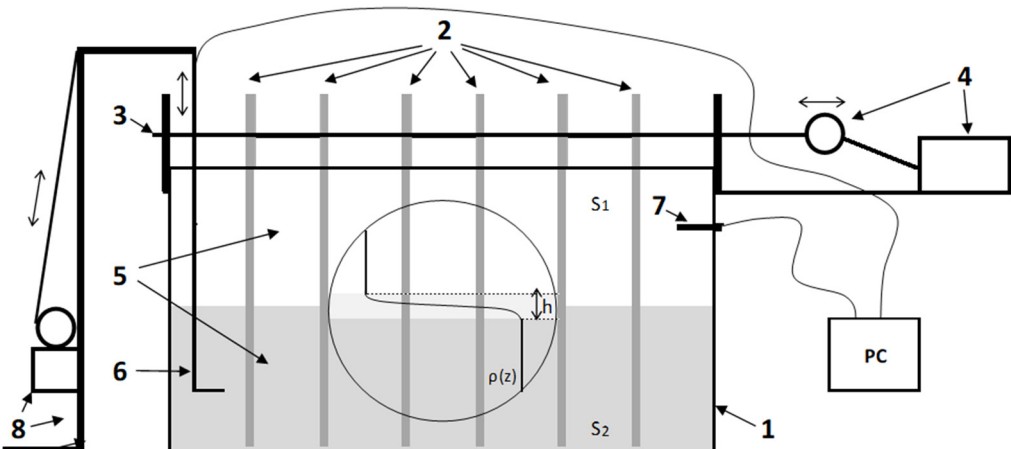

**Figure 2.** Experimental setup: **1**, an organic glass tank; **2**, a grid system with vertical glass rods; **3**, a metallic rod on which six grids with the glass rods are fixed; **4**, an electric motor with an eccentric providing horizontal oscillation of the metallic rod; **5**, two-layered fluid with an intermediate layer of a high density (salinity) gradient; **6**, single-electrode conductivity microsensor for measuring the salinity profile; **7**, four-electrode conductivity sensor for measuring salinity in the upper quasi-uniform layer; PC, personal computer for registration and processing of measured data; $S_1$ and $S_2$, salinity of the upper and lower quasi-uniform layers; **8**, an elevator, which provides vertical displacement of the conductivity microsensor. The circle in the center of the tank indicates the section of the plane-parallel beam of light by the shadowgraph device.

Before each experimental run, the tank was half filled with fresh distilled water, below which a water solution of NaCl of a given concentration was poured slowly to avoid mixing. As a result, a two-layered medium with layers of different salinity and equal thickness ($H_1 = H_2$, where $H_1$ is the thickness of the upper layer, and $H_2$ is the thickness of the lower one) was formed with a narrow density gradient zone $h \ll H_1, H_2$ between the layers. The temperature of the layers was the same with the accuracy of $\pm 0.2$ °C and close to room temperature.

After the tank was completely filled with two-layered water medium, the stirring with oscillating rods was switched on and this moment in time was taken as a start of an experimental run. An eccentric driven by the motor provided a sinusoidal oscillation of the metallic rod with the grids of glass rods for a given amplitude and period. During each experimental run, the current salinity $S_1$ in the upper layer was measured using a laboratory conductivity meter Expert 002. It is a four-electrode sensor with a diameter 0.5 cm, and with a length 3.0 cm. Using six subranges it provides the measurement of electrical conductivity in a range from 0.001 μS/cm up to 1000 mS/cm. The standard error

of conductivity measurements does not exceed 2% (https://magazinlab.ru/konduktometr-jekspert-002.html, accessed on 25 May 2022).

A value of $S_1$ gradually increased with time due to the flux of salt from the lower layer across the density gradient layer. Throughout each experiment the thickness of the upper and lower layers remained practically unchanged. The four-electrode sensor of the conductivity meter Expert 002 was placed horizontally in the upper layer at a distance of 1.5 cm below the water surface in the tank. The measurements of the electrical conductivity with the four-electrode sensor were carried out with an interval of 1 s from the beginning to the end of the experimental run.

In all of experiments, along with the measurements of the salinity in the upper layer with the conductivity meter Expert 002, the vertical profiles of conductivity were measured using a single-electrode conductivity microsensor. It was attached to a mechanical yo-yo device providing vertical movement of the sensor at a speed of about 0.2 cm s$^{-1}$. Since the diameter of the electrode made from nichrome wire was only 0.01 cm, its vertical resolution was about of 0.1 cm. The signal registration frequency was 10 Hz. This made it possible to obtain almost non-smoothed vertical structure of the salinity distribution in the upper and lower layers and across the density gradient layer.

It should be noted that before each series of the experiments, the conductivity sensors were calibrated by immersing them together into plastic glasses filled with water of a given salinity. Initially, the salinity of the water in the glasses was set by the gravimetric method, and then it was determined with an accuracy of 0.01 ppm using the laboratory salinometer Autolab (Manly, Australia). The error of conductivity measurements using the conductivity meter Expert 002 did not exceed 2% of the measured value. The data of conductivity measurements by a single-electrode microsensor in the upper layer were referenced to the simultaneous readings of the conductivity meter Expert 002. The stability of salinity measurements with a single electrode microsensor was quite high; its drift did not exceed 0.1 ppm per day. In addition to the quantitative measurements described above, the Schlieren shadowgraph device was used to produce the picture of a density structure and fluctuations in the stratified fluid at the screen behind the tank (Figure 2). The shadowgraph images were photographed regularly during the experiments and compared with the salinity profiles measured at the same time when it was possible. Due to the use of a shadowgraph method, it was possible to monitor the density interface position even in the absence of measurements of the vertical salinity profiles by a single-electrode conductivity microsensor.

The laboratory study consisted of 39 experimental runs most of which were carried out at different values of the defining dimensional and dimensionless parameters (see Table 1). A full description of these options is provided in the next section of the paper. The experiment was carried out during the time period from June 2021 to February 2022.

**Table 1.** The values of the defining dimensional and dimensionless parameters of the experiment: $\Delta S^{initial}$, the initial value of the salinity difference between the layers; $Re$, the Reynolds number; $Ri^{initial}$, the initial value of the Richardson number; $2A$, double amplitude of the rods oscillation; $T$, the period of the oscillations of the rods.

| Experiment | $\Delta S^{initial}$, ppm | $Re$ | $Ri^{initial}$ | $2A$, cm | $T$, s |
|------------|---------------------------|------|----------------|----------|--------|
| 1 | 10 | 135 | 84 | 2.6 | 4.0 |
| 2 | 10 | 132 | 89 | 3.3 | 5.2 |
| 3 | 10 | 134 | 86 | 3.3 | 5.1 |
| 4 | 12 | 147 | 85 | 3.3 | 4.6 |
| 5 | 14 | 159 | 84 | 3.1 | 4.0 |
| 6 | 14 | 160 | 84 | 3.3 | 4.3 |

**Table 1.** *Cont.*

| Experiment | $\Delta S^{initial}$, ppm | *Re* | $Ri^{initial}$ | 2*A*, cm | *T*, s |
|---|---|---|---|---|---|
| 7 | 14 | 159 | 86 | 3.3 | 4.3 |
| 8 | 14 | 158 | 86 | 3.3 | 4.3 |
| 9 | 16 | 170 | 85 | 3.3 | 4.0 |
| 10 | 18 | 180 | 86 | 3.3 | 3.8 |
| 11 | 19 | 168 | 102 | 3.3 | 4.1 |
| 12 | 20 | 190 | 84 | 3.7 | 4.0 |
| 13 | 20 | 194 | 81 | 3.3 | 3.5 |
| 14 | 24 | 207 | 86 | 3.3 | 3.3 |
| 15 | 24 | 211 | 83 | 3.3 | 3.2 |
| 16 | 24 | 207 | 86 | 3.3 | 3.3 |
| 17 | 24 | 204 | 89 | 3.3 | 3.3 |
| 18 | 24 | 204 | 89 | 3.3 | 3.3 |
| 19 | 28 | 223 | 86 | 3.3 | 3.1 |
| 20 | 28 | 223 | 87 | 3.3 | 3.1 |
| 21 | 30 | 214 | 100 | 3.3 | 3.2 |
| 22 | 30 | 214 | 100 | 3.3 | 3.2 |
| 23 | 30 | 214 | 100 | 3.3 | 3.2 |
| 24 | 32 | 238 | 87 | 3.3 | 2.9 |
| 25 | 36 | 254 | 86 | 3.3 | 2.7 |
| 26 | 38 | 238 | 103 | 3.3 | 2.9 |
| 27 | 42 | 274 | 86 | 3.3 | 2.5 |
| 28 | 46 | 287 | 86 | 3.3 | 2.4 |
| 29 | 46 | 288 | 85 | 3.3 | 2.4 |
| 30 | 48 | 267 | 104 | 3.3 | 2.6 |
| 31 | 56 | 293 | 96 | 3.3 | 2.3 |
| 32 | 56 | 317 | 86 | 3.3 | 2.2 |
| 33 | 64 | 341 | 84 | 3.3 | 2.0 |
| 34 | 75 | 364 | 87 | 3.3 | 1.9 |
| 35 | 76 | 370 | 85 | 3.3 | 1.8 |
| 36 | 88 | 365 | 102 | 3.3 | 1.9 |
| 37 | 90 | 402 | 86 | 3.3 | 1.7 |
| 38 | 108 | 441 | 85 | 3.3 | 1.5 |
| 39 | 139 | 498 | 86 | 3.3 | 1.4 |

## 3. Results

### 3.1. Dependence of Entrainment Velocity on Richardson Number

During every experimental run the stirring of the two-layer water medium using oscillating vertical rods was carried out from initial state until complete mixing of the layers (i.e., until achieving the homogeneity of the water column). The buoyancy flux $F_b$ between the layers was calculated using the time series of the salinity values in the upper layer:

$$F_b = g\beta(dS_1/dt)H_1 \tag{1}$$

where $S_1$ is the instant salinity of the upper layer, $t$ is the time, $dS_1/dt$ is the rate of the salinity change in the upper layer, $g$ is the acceleration of gravity, $\beta$ is coefficient of salinity compression. Neglecting the thickness $h$ of the density transition zone between the upper and lower layers, one can rewrite the buoyancy flux between the upper and lower quasi-homogeneous layers in the form:

$$F_b = g\beta U_e \Delta S \tag{2}$$

where $U_e$ is the rate of turbulent entrainment of the water from the lower layer to the upper layer, and $\Delta S = S_2 - S_1$ is the instant salinity difference across the density interface between the layers, and $S_2 = S_2^{initial} - S_1$. Therefore, $\Delta S = S_2^{initial} - 2S_1$, where $S_2^{initial}$ is the initial salinity value of the lower layer. Equating the right-hand sides of (1) and (2), we obtain an expression for $U_e$:

$$U_e = (dS_1/dt)H_1/\Delta S = (dS_1/dt)H_1/\left(S_2^{initial} - 2S_1\right) \tag{3}$$

Following [6–8] the experimental data were presented in dimensionless form and analyzed based on the dimensionless Richardson and Reynolds numbers. The dimensionless form of $U_e$ was obtained by dividing it on the velocity scale to some extent corresponding to the velocity scale of energy-carrying eddies in the turbulent layers.

In [7,8], this velocity scale was taken as an average speed of sinusoidal oscillation of the rods $U = 4A/T$, where $A$ is the amplitude and $T$ is the period of the rod oscillations. However, the zone of turbulence generation occupied by the oscillating rods engages a minor part of the horizontal section of the tank. Therefore, we came to conclusion that for the scale of the velocity of turbulent fluctuations caused by energy-carrying vortices, it is reasonable to choose the normalized velocity of the rod oscillations, which can be expressed as:

$$U = 4AT \cdot (\Sigma_r/\Sigma_0) \tag{4}$$

Here $\Sigma_r = NB2dA$ is the turbulence generation horizontal area, $N$ is the number of vertical rods in the tank, $\Sigma_0$ is the area of the horizontal section of the tank, and $B$ is an empirical coefficient that should be greater than one ($B > 1$). The basis for this assumption is the fact that the area $\Sigma_r$ should be larger than the valence $2Ad$ due to presence of the viscous boundary layer around the rod and the mass of fluid dynamically attached to it. Taking into account that the efficiency of turbulent mixing, which is the fraction of the kinetic energy of turbulence spent on increasing the potential energy of a stratified fluid, reaches a maximum of about 20% [12,14], we obtained that the value of the coefficient $B$ is approximately equal to 2.4, which seems to be rather realistic. The justification for such a parametrization of the turbulence velocity scale, as well as the length scale of energy-containing turbulent pulsations (eddy diameter) $l \approx M$, where $M$ is the mesh size of the grid will be discussed in Section 3.5.

Based on the chosen scales of turbulent fluctuations in mixed layers, one can introduce two key dimensionless parameters of the experiment as follows: the Reynolds number, $Re = UM/\nu$, where $\nu$ is the kinematic viscosity of water, and the Richardson number, $Ri = g\beta\Delta SM/U^2$.

In the logarithmic coordinates, the dependence of $U_e/U$ on the $Ri$ number makes possible to verify the concept of the power-law dependence of the dimensionless entrainment rate on the Richardson number in a two-layered stratified aqueous medium [12]:

$$U_e/U = CRi^{-n} \tag{5}$$

Here $n$ is an exponent to be determined as a result of the experiments. In all experimental runs, the behavior of the experimental points approximated by (5) was similar since there existed two pronounced quasi-linear parts in the plot (Figure 3).

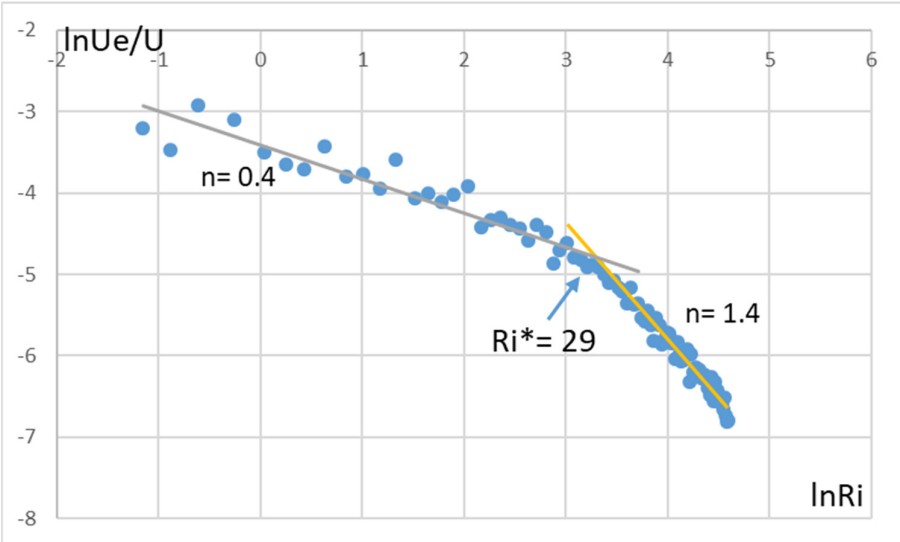

**Figure 3.** Dependence of $\ln U_e/U$ on $\ln Ri$ for the experimental run with the initial value of $Ri^{initial} = 100$ and $Re = 214$. The straight lines show two different asymptotic power dependences. Their intersection point at $Ri^*$ indicates to the abrupt transition from one dependence to the other.

At vicinity of the point of intersection of straight lines at Figure 3 there is an abrupt transition from one turbulent mass transfer mode to the other. It should be noted that both values of the exponent $n$ and the coefficient $C$ in (5), as well as the value of $Ri^*$, are the functions of the Reynolds number $Re$. More detailed information on the dependences of $C$, $n$, and $Ri^*$ on $Re$ for $Ri > Ri^*$ will be given in Section 3.4.

*3.2. Structure of the Density Transition Layer*

A pattern of time evolution of the salinity profiles was obtained in the experimental runs. Since each experimental run began with a certain maximum value of $Ri$ (with the greatest difference in salinity (density) between the layers), and ended when $Ri = 0$ (no density difference between the layers), it was possible to obtain the dependence of the density transition zone thickness and its structure on the Richardson number. In a wide range of its variation, the behavior of the density interface was found to be as follows. At the beginning, the thickness of the interface zone after a rather short stage of its adaptation to the turbulent stirring maintained in a sharpened state (Figure 4a–f) and then at $Ri \approx Ri^*$ it starts to expand (Figure 4g,h). These findings qualitatively agreed with the results of the experiment in which the density transition zone was formed under the impact of turbulence generated by vertically oscillating horizontal grids in the upper and lower water layers of different salinity and temperature [13].

Figure 5 shows the typical vertical salinity structure of the fluid column divided into three sublayers: two quasi-homogeneous sublayers (top and bottom) with thicknesses $h_1$ and $h_3$, correspondingly, and a middle sublayer with a maximum salinity gradient (density interface) with a thickness $h_2$. The thickness $h_2$ was determined as the vertical distance between the points of separation of the tangents from the real profiles.

Figure 6 shows an evolution of $h_1$ and $h_2$ with time in details, taking into account all of the salinity profiles obtained during the experimental run.

It follows from Figures 5 and 6 that $h_2$ characterize the thickness of the sublayer of the maximum salinity gradient (density interface) displaced between the upper and lower quasi-homogeneous sublayers. The density interface plays a key role in suppressing turbulence generated by the oscillating grids of vertical rods and preventing the penetration of eddies from one quasi-homogeneous sublayer to the other. Therefore, the characteristics of this sublayer most clearly reflect the change in the regime of mass transfer between the layers when the critical value of the Richardson number is reached.

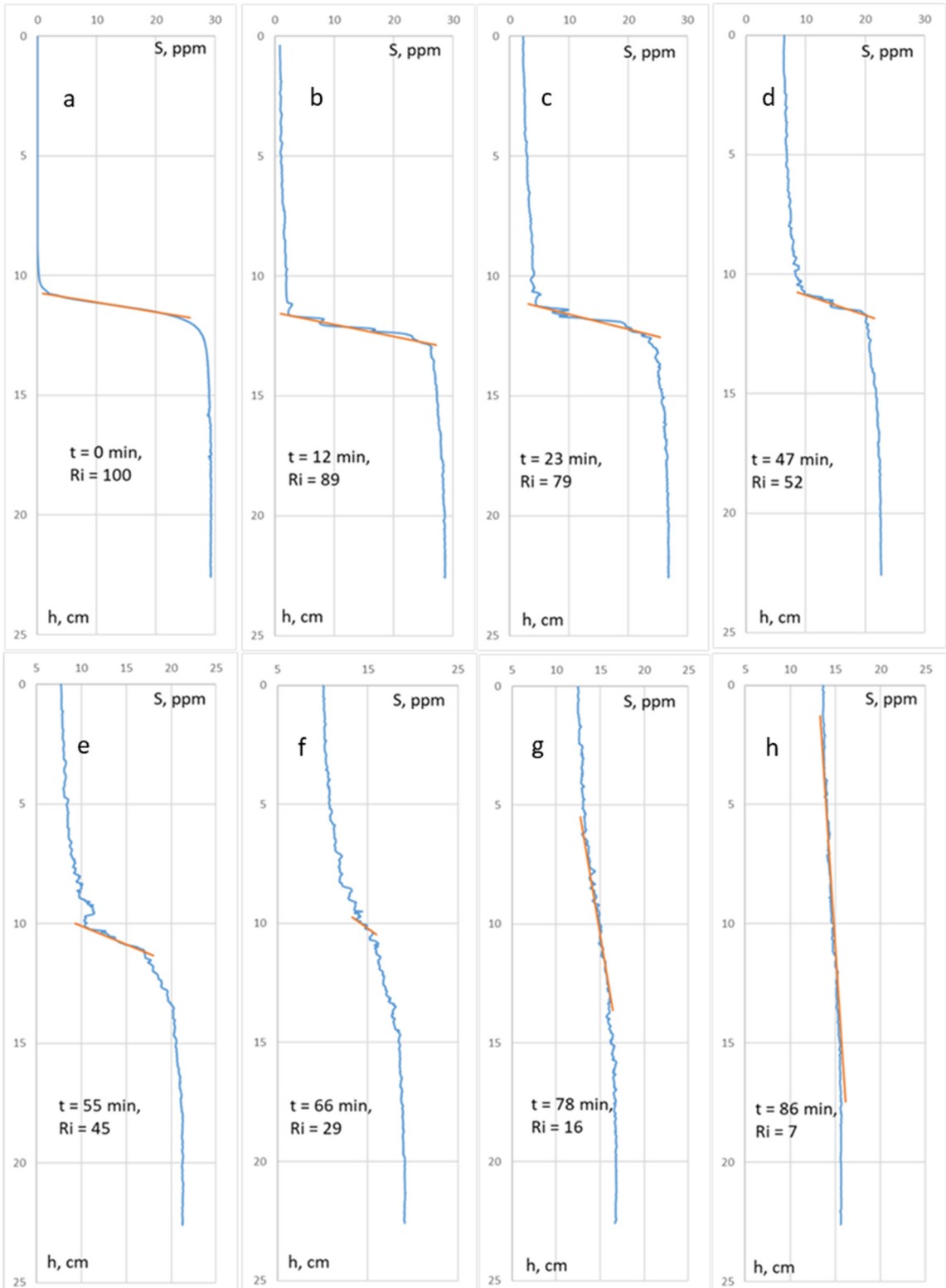

**Figure 4.** The sequence of salinity profiles measured during one of the experimental runs with initial conditions as follows: $\Delta S^{initial} = 30$ ppm, $Ri^{initial} = 100$, and $Re = 214$. Tangent lines (marked in orange) are added to the quasi-linear sections of the profiles with the maximum salinity gradient. The subfigures are shown during the experimental run at the time and the Richardson number as follows: (**a**): $t = 0$ min, $Ri = 100$; (**b**) $t = 12$ min, $Ri = 89$; (**c**) $t = 23$ min, $Ri = 79$; (**d**) $t = 47$ min, $Ri = 52$; (**e**) $t = 55$ min, $Ri = 45$; (**f**) $t = 66$ min, $Ri = 29$; (**g**) $t = 78$ min, $Ri = 16$; (**h**) $t = 86$ min, $Ri = 7$.

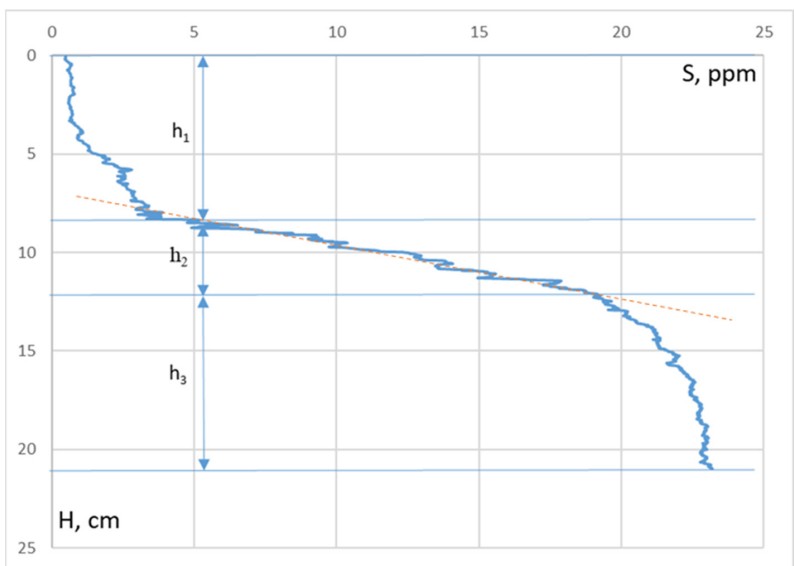

**Figure 5.** Vertical distribution of salinity in a two-layer fluid with an $\Delta S = 23$ ppm, $Ri = 86$ and $Re = 204$. Tangent line (marked in orange) is added to the quasi-linear sections of the profiles with the maximum salinity gradient.

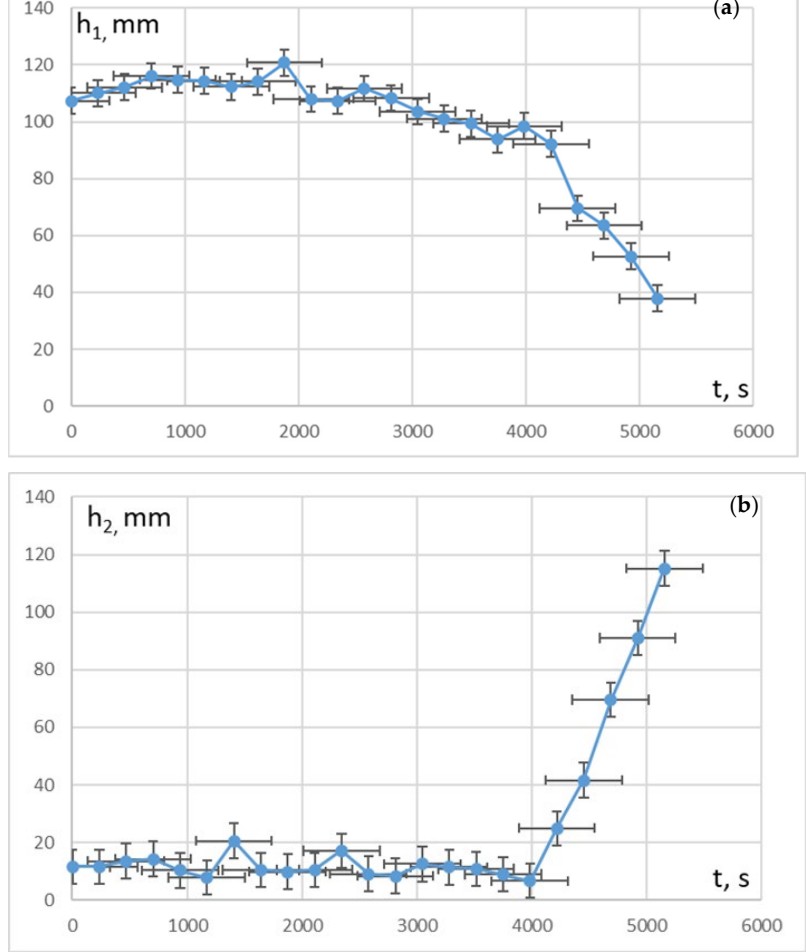

**Figure 6.** Variations of the sublayers thickness versus time: (**a**) the quasi-homogeneous sublayer thickness $h_1$; (**b**) the salinity interface sublayer thickness $h_2$, $Ri^{initial} = 100$; $Re = 214$. The bars display the standard deviation of the data.

The point at which the derivative of the $h_2/M(Ri)$ function change characterizes the transition from the mode of the density interface "sharpening" to the mode of its "erosion" (turbulent diffusion). In Figure 7 this transition corresponds to the number $Ri = Ri^* = 29 \pm 7$ at $Re = 214$. This result support a conclusion that during a vertically homogeneous turbulent stirring of the initially continuously stratified fluid, the fine-structure layering (step-like structure formation) is possible only at $Ri > Ri^*$.

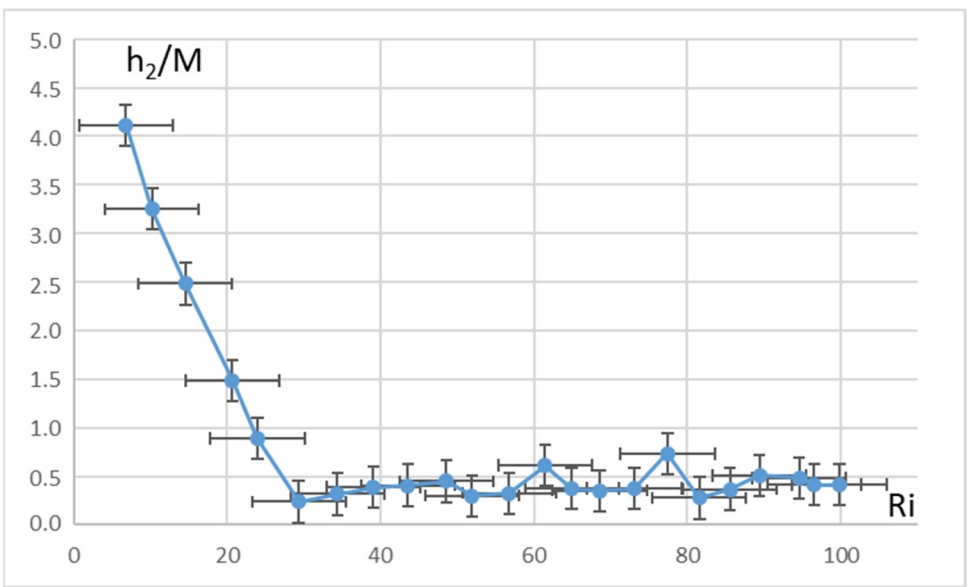

**Figure 7.** Dependence of $h_2/M$ on $Ri$ for the experimental run with initial salinity difference $\Delta S = 30$ ppm, $Ri^{initial} = 100$, and $Re = 214$. The bars are standard errors of measurements.

It should be noted that at $Ri < Ri^*$ the dimensionless thickness of the density interface is not only a function of $Ri$ and $Re$, but also of time. Thus, it can be assumed that at $Ri < Ri^*$ the density boundary exists in a non-stationary regime and tends to expand over the entire thickness of the fluid layer in the tank with time. On the contrary, for $Ri > Ri^*$ the density interface exists in a quasi-stationary self-sustaining regime.

### 3.3. Mixing Efficience and It Dependence on Richardson Number

It is important to analyze the dependence of the Richardson flux number $Rf$ as a function of $Ri$. By its physical meaning, $Rf$ is the fraction of the kinetic energy of turbulence spent on mixing (i.e., on increasing the potential energy of a stratified fluid system) [12]. Here,

$$Rf = (U_e/U)Ri \qquad (6)$$

The dependence of $Rf$ on $Ri$ (Figure 8), reaches a maximum value close to 0.2 at $Ri = Ri^* \approx 29$, when $Re = 214$. The maximum corresponds well to the point where the derivation of $h_2/H_1(Ri)$ change (Figure 7). Thus, it characterizes the transition from the sharpening interface mode to an eroding mode.

### 3.4. Dependence on the Reynolds Number

The experiments revealed a significant dependence of both $C$ and $n$ on $Re$ at $Ri > Ri^*$ (see Formula (5)). These functions are shown in Figures 9 and 10, and by expressions (7) and (8), respectively.

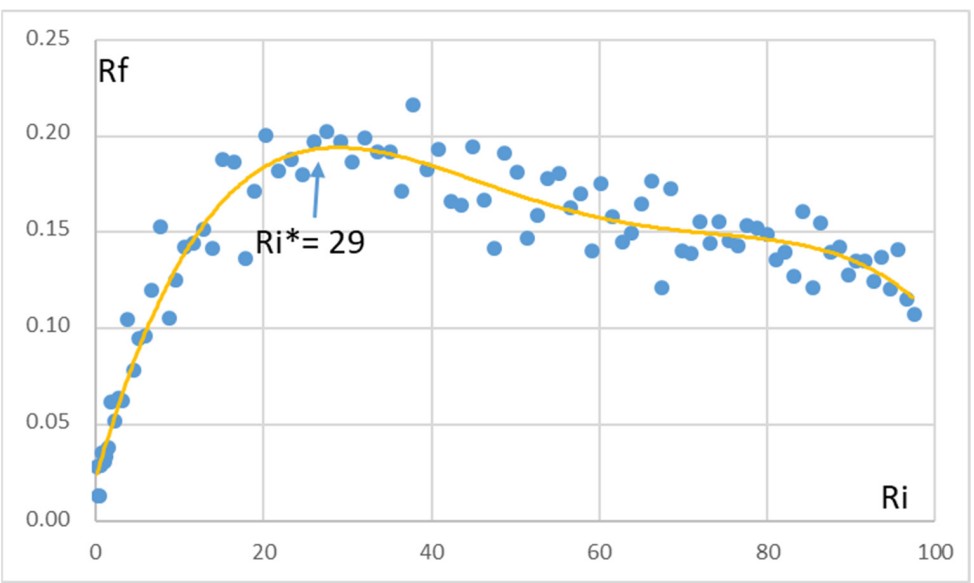

**Figure 8.** Dependence of $Rf$ on $Ri$ for the experimental run with $Ri^{initial} = 100$ and $Re = 214$.

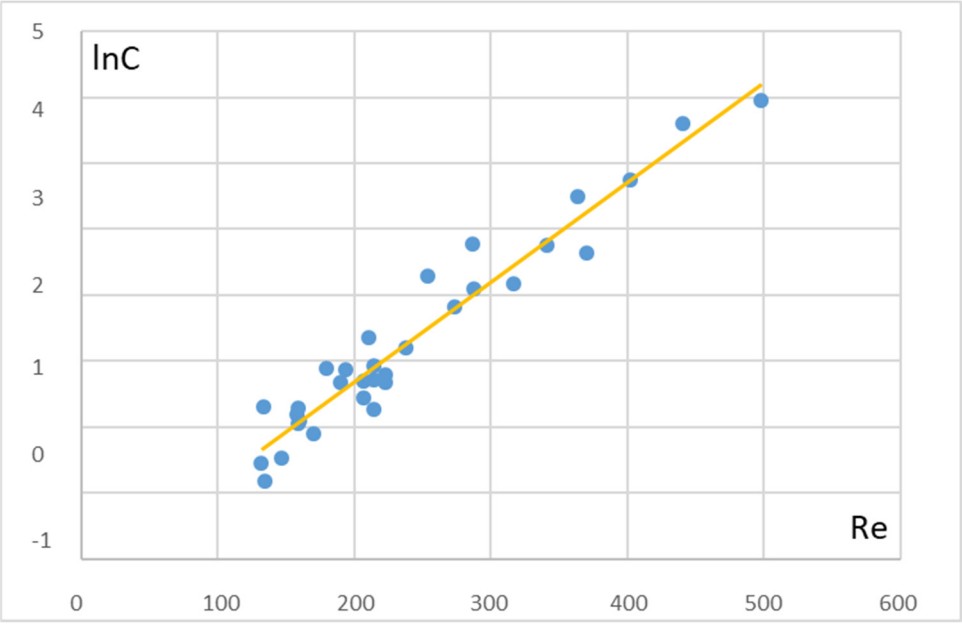

**Figure 9.** The dependence of $\ln C$ on $Re$ for the case $Ri > Ri^*$.

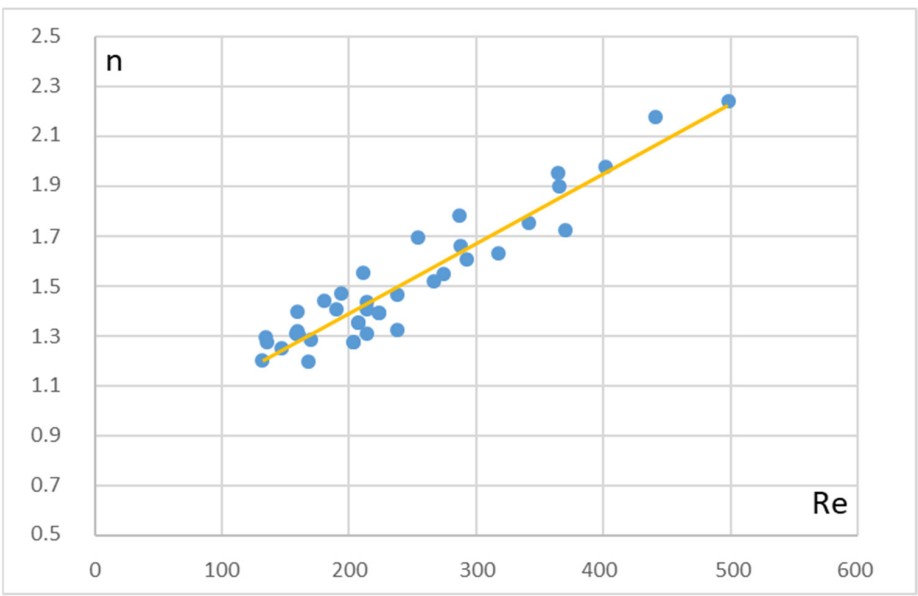

**Figure 10.** The function $n(Re)$ for the case $Ri > Ri^*$.

As follows from Figure 9, there was a quasi-linear dependence of $\ln C$ on $Re$ given by

$$\ln C = 0.02Re - 3.33 \tag{7}$$

There was also a quasi-linear dependence of the exponent $n$ on $Re$ for the case $Ri > Ri^*$ as follows:

$$n(Re) = 0.003Re + 0.87 \tag{8}$$

As for the dependences of $C$ and $n$ on $Re$ at $Ri < Ri^*$, they also exist, but their study is problematic, since in the mode of erosion of the density interface, the regularities of vertical turbulent mass transfer are also a function of time. Due to these circumstances, we did not consider it relevant to determine these dependencies.

At the critical Richardson number $Ri^*$, the regularities of the mass transfer between turbulent layers were changed. A dependence of $Ri^*$ on the Reynolds number $Re$ (Figure 11) was also observed.

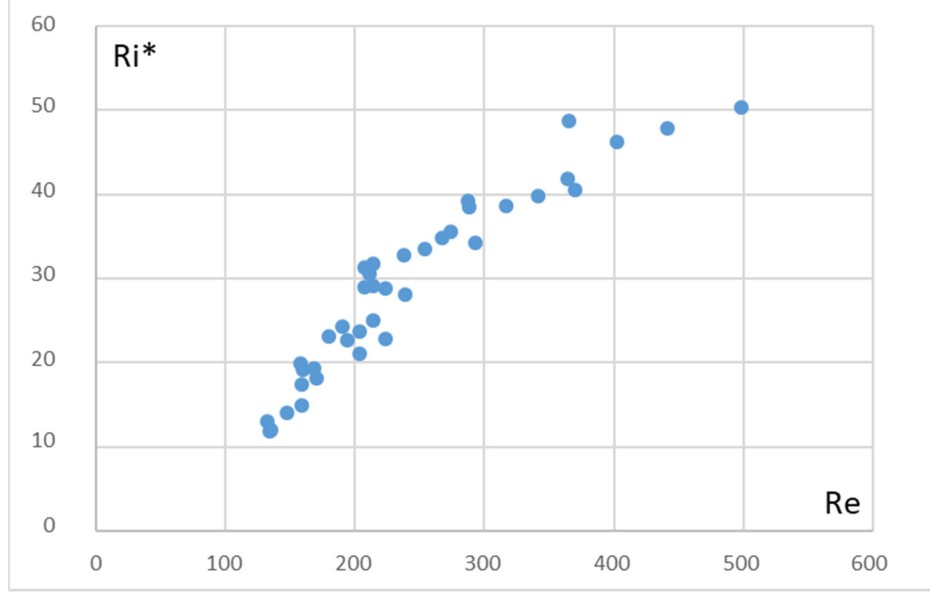

**Figure 11.** The critical Richardson number $Ri^*$ as a function of the Reynolds number $Re$.

It follows from Figure 11 that $Ri^*$ is a growing function of $Re$ at least at low values of this parameter. An increase in the value of the critical Richardson number with the Reynolds number at which the mode of expansion of the density interface between layers is replaced by the mode of its sharpening was found previously in laboratory experiments on stirring of linearly stratified water medium [6,7]. However, there is a reason to expect that the increase of $Ri^*$ slows down with increasing $Re$ and it is possible that the function $Ri^*(Re)$ has a finite limit.

### 3.5. Analysis of Salinity Pulsations at the Density Interface and an Estimate of Energy-Containing Eddies Scale

Some useful information about the process of turbulent exchange between the layers of different density can be obtained from the analysis of salinity fluctuation records measured by electrical conductivity microsensors. For this purpose, two special experimental runs were carried out. In one of them, along with registration of shadowgraph images of the interface at different moments of time and measuring salinity profiles with an electrical conductivity microsensor, another conductivity microsensor was fixed inside the density interface to measure salinity fluctuations caused by turbulent mixing.

The salinity profile measured by the conductivity microsensor at one of the moments of time during the density interface sharpening mode is presented at Figure 12. The interface was very narrow: its thickness was about 1 cm and the value of the Brent-Väisälä frequency $N \approx 4\,\text{s}^{-1}$. This means that the period of free density interface oscillations $T \approx 1.6$ s was twice less than the period of vertical glass rods oscillation $T = 3.2$ s.

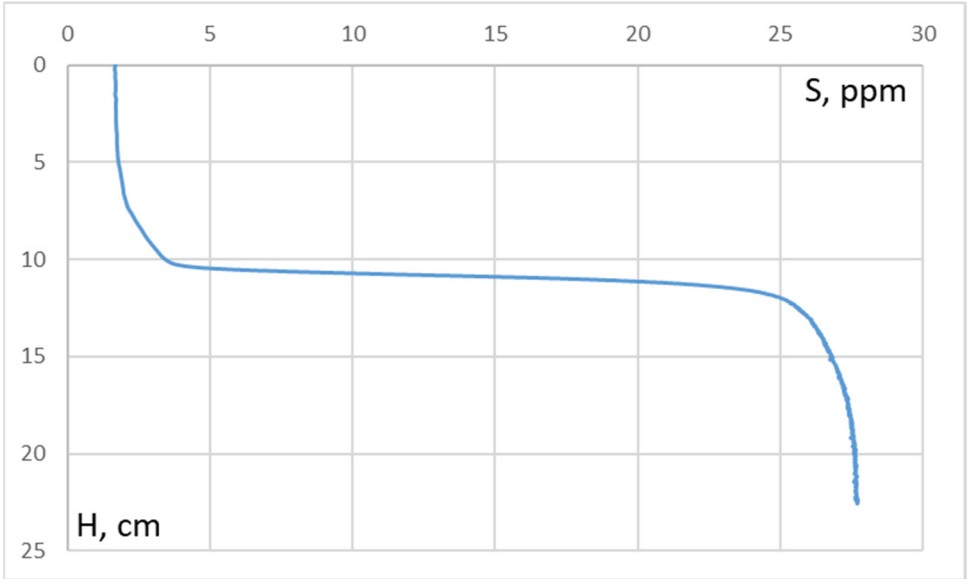

**Figure 12.** The snapshot of the salinity profile measured by the conductivity microsensor at a moment of time during the density interface sharpening ($Ri^{initial} = 100$; $Re = 214$ ).

The shadowgraph image of the density interface received nearly at the same time as the salinity profile at Figure 12 is presented at Figure 13. This image shows undulating curvature of the interface caused (possibly) by the impact on it of rather large (several centimeters in size) eddies that exist in the upper and lower layers together with the interfacial internal waves.

The readings of the electrical conductivity sensor located in the area of the density interface, is shown at Figure 14. Here, the red line is the smoothed data of salinity during the experimental run, and the blue line is the raw data indicating high-amplitude salinity fluctuations in a wide frequency range. An ensemble of the data shown on the graph by a double-sided arrow was taken for spectral analysis after removal of the linear trend.

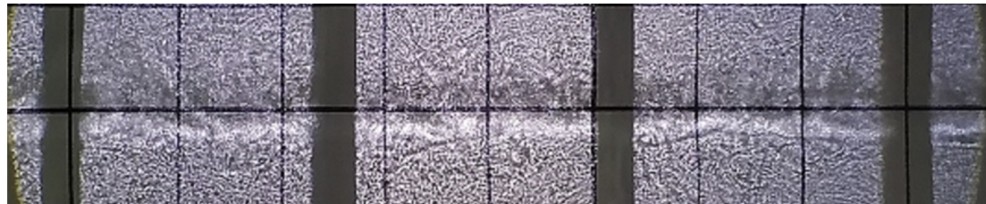

**Figure 13.** The shadowgraph image of the density interface received nearly at the same time as the salinity profile at Figure 12 ($Ri^{initial} = 100$; $Re = 214$ ).

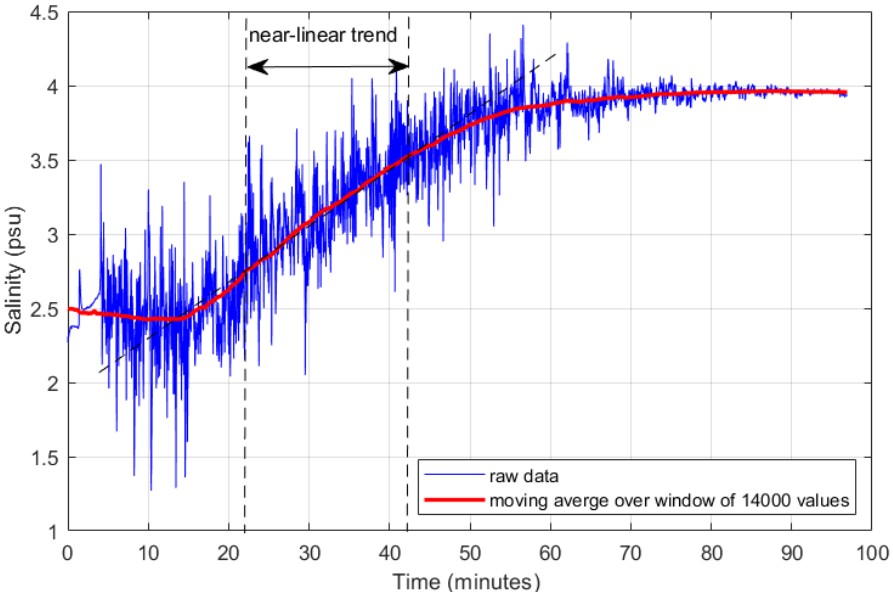

**Figure 14.** The salinity (blue line) as derived from the conductivity data registered by the microsensor of electrical conductivity with a frequency of 10 Hz in the density interface layer. Red line indicates moving average of the salinity. The vertical dashed bars indicate the data ensemble used for the spectral analysis.

The power spectrum of salinity fluctuations presented at Figure 15 was rather smooth without pronounced peaks. This was surprising since, in our opinion, the frequency of the rods oscillation, as well as its overtones, should have appeared. One of the possible reasons for the absence of these peaks, associated with the oscillation frequency of the rods, may be the mutual influence of neighboring grids of rods on turbulent fluctuations, causing a masking effect.

To test this hypothesis, an experimental run was carried out where only one grid with rods (instead of six grids) was oscillating in a tank while the other conditions were the same. As before, an electrical conductivity microsensor was located near the oscillating grid in the area of the density interface between the layers. The spectral energy of the measured salinity fluctuations is shown at Figure 16. The peaks of energy at $f = 3.3 \times 10^{-1}$ s$^{-1}$ and $6.6 \times 10^{-1}$ s$^{-1}$ were clear-cut. These frequencies corresponded well to the grid oscillation period of 3.2 s and to its double frequency overtone. This confirmed the hypothesis that the absence of peaks associated with the oscillation frequency of the rods at Figure 14 was related to the mutual influence of neighboring grids of rods on the spectral composition of turbulent fluctuations. As for the threshold frequency at which the spectral energy reached its maximum values and a plateau (see Figure 15), it was about $3 \times 10^{-2}$ s$^{-1}$ which corresponded to the time scale $\tau \approx 30$ s. If we assume that this time scale characterized the overturn period of fluid particles in energy-carrying turbulent eddies, and the particle velocity $U$ was determined by Formula (4), it was easy to obtain that $U\tau \approx \pi M$. Thus, the spectral analysis of salinity fluctuations indirectly confirmed the validity of the velocity scale $U$ parametrization by Formula (4), as well as the fact that eddy diameter $l$ was close

to *M* i.e., to the mesh size of the grid. Still, it is necessary to refine these scales on the basis of direct measurements of turbulence parameters, which would be the task for the future studies.

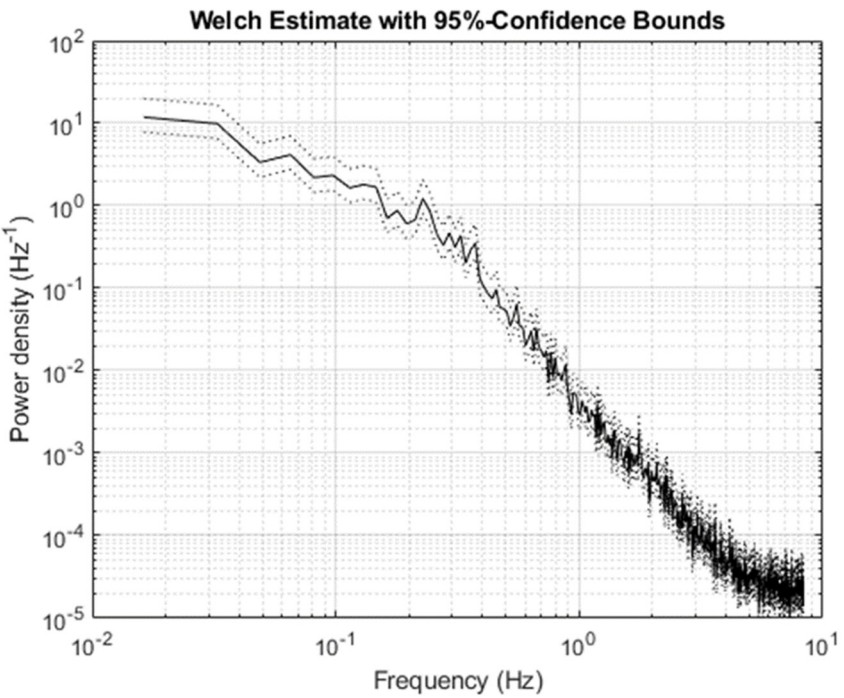

**Figure 15.** The variance-normalized power spectral density of salinity fluctuations shown in Figure 14 ($Ri^{initial} = 100$, $Re = 214$ ).

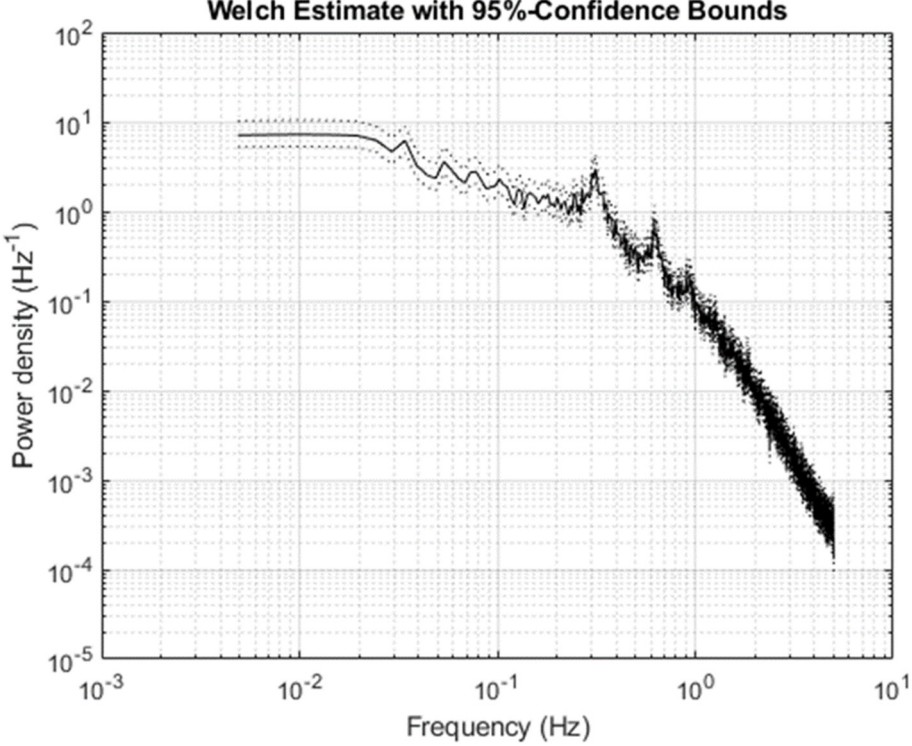

**Figure 16.** The variance-normalized power spectral density of the salinity fluctuations obtained in the experiment with the single oscillating grid.

*3.6. Density Interface Structure, Mass Flux between the Turbulent Layers and Their Relation with Phillips-Posmentier Mechanism*

The results of the experimental runs described above confirmed that the sharpening of the density interface between the turbulent layers of different salinity occurred in accordance with the Phillips-Posmentier model [3,4] as follows. In a turbulent stratified shear flow, the vertical mass flux can be represented as:

$$F = K(d\rho/dz), \tag{9}$$

where $K$ is coefficient of turbulent exchange, $d\rho/dz$ is the vertical density gradient. The vertical exchange in a stratified fluid depends on the shear Richardson number $Ri_s = g(d\rho/dz)/\rho(dU/dz)^2$ [15], where $dU/dz$ is the velocity shear. The coefficient $K$ is represented by a decreasing power function of the Richardson number

$$K \sim C_1 Ri_s^{-n} \tag{10}$$

If $n > 1$, the mass flux $F$ (9) is a decreasing function of the density gradient $d\rho/dz$. This means that if somewhere in the flow the density gradient increases locally then the mass flux through this region decreases. As a result, the gradient should increase further.

This situation is dynamically unstable, since a small deviation from equilibrium leads to a further increase in the deviation. Stratification tends to break up into turbulent layers of uniform density, separated by narrow interfaces with a large density gradient. For $n \leq 1$, such instability is absent, since the local inhomogeneities of the density gradient degenerate under the impact of the vertical mass flux.

Dealing with the obtained experimental data, we can move from describing turbulent mass transfer through the density interface via the entrainment velocity $U_e$ to vertical turbulent exchange coefficient $K$. In our case, the integral estimate of turbulent exchange coefficient in the upper layer could be expressed as $K = U_e \cdot h_1$. Since both $U_e$ and $h_1$ were measured in the experimental runs, it is possible to construct the dependence $K(Ri)$, or $Nu(Ri)$, where $Nu = K/k$ is the Nusselt number for salt and $k = 1.3 \times 10^{-5}$ cm$^2$s$^{-1}$ is the salt molecular diffusion coefficient. The dependence $Nu(Ri)$ is expressed by:

$$Nu \sim C_2 Ri^{-n} \tag{11}$$

A plot of $\ln Nu$ versus $\ln Ri$ is shown in Figure 17. It follows from the figure that in the mode of the density interface sharpening ($Ri > Ri^*$), the value of the exponent in the Formula (11) $n = 1.4$ (>1) and at $Ri > Ri^*$, $n = 0.2$ (<1). It is obvious that these values of $n$ agree well with the Phillips-Posmentier model. It should be mentioned that the molecular flux of salt in this experiment is much lower than the turbulent one even during the mode of interface sharpening ($Ri > Ri^*$).

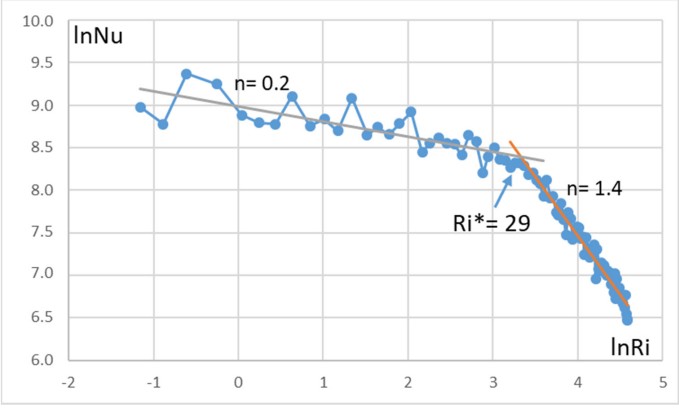

**Figure 17.** Dependence of $\ln Nu$ on $\ln Ri$ in the experimental run with $\Delta S^{initial} = 30$ ppm and $Re = 214$. $C_2 = 13$ for the asymptote at $Ri > Ri^*$.

## 4. Discussion

As it was shown in the Section 3.3, at $Ri \approx Ri^*$ the efficiency of the turbulent kinetic energy transition into the potential energy of stratification, reaches a maximum value, which is about 10–30%. Linden [12,14] analyzed the results of his experiments, as well as those carried out by other authors, and arrived at conclusions quite close to that. This means that we have chosen velocity and length scales of the energy-bearing turbulent eddies close to the real ones. At the same time, it is necessary to investigate how these parameters of turbulence are related to the parameters of vertical rods oscillation. Only after that will it be possible to apply findings of the laboratory experiment to natural conditions.

Of course, it can be argued that shear-free grid turbulence has nothing to do with natural conditions where turbulent mixing occurs in the regions of flow velocity shear, which is often observed at the density interfaces. In particular, Peligri and Sandra [15], proposed an alternative to Phillips-Posmentier model of mixing in stratified shear flow. According to [15], the maximum vertical mass flux occurs in the area of the maximum vertical density gradient, and was absent in the area of low-density gradient. As a result, a mixed layer is formed in the region of the initial density interface, whereas new density interfaces are formed above and below the mixed layer. For an experimental confirmation of this mechanism, Peligri and Sandra [15] referred to the experiment of Thorpe [16] conducted in an inclined sealed pipe filled with a two-layer stratified miscible fluid. As a result of the formation of a chain of turbulent Kelvin-Helmholtz billows along the length of the pipe and the mixing caused by them, the interface splits at two new density interfaces formed above and below the initial interface. Implementation of this process in natural conditions requires at least two conditions. Firstly, the region of development of the Kelvin-Helmholtz instability should have a large horizontal extension in order not to be collapsed by the stratification. Secondly, turbulence should persist for a long time to form a well-mixed region bounded above and below by sharp density interfaces. However, it is unclear so far how often such conditions can be realized in nature.

The close type of criticism on the realization of the Phillips-Posmentier mechanism was presented in [17]. The study concerned with the mixing in laboratory experiments in which an initially linearly stratified fluid was stirred with a rake of vertical bars. It was found that the flow evolution strongly depended on the Richardson number *Ri*. At low *Ri*, the buoyancy flux was a function of the local buoyancy gradient only, and could be modelled as gradient diffusion with a *Ri*-dependent eddy diffusivity. At high *Ri*, vertical vorticity shed in the wakes of the bars interacted with the stratification and produced well-mixed layers separated by interfaces. This process leads to layers with a thickness proportional to the ratio of grid velocity to buoyancy frequency for a wide range of Reynolds numbers *Re* and grid solidities. In this regime, the buoyancy flux was not a function of the local gradient alone, but also depended on the local structure of the buoyancy profile. It was proposed that, the layers were not formed by the Phillips-Posmentier model, but they resulted from vortical mixing previously thought to occur only at low *Re*.

The important factor of the impact of molecular effects on the turbulent diffusivity [18] and step-like structure formation in stratified shear flow was investigated in [11]. In particular, direct numerical simulation was used to investigate the process dependence on three non-dimensional parameters as follows the gradient Richardson number, the buoyancy Reynolds number and the Prandtl number. A simple expression for conditions favorable for the layer formation was obtained. It was shown that the layers were formed only in the case of rather large values of Prandtl number.

## 5. Conclusions

Overall, we would like to make two preliminary conclusions related to the regularities of the evolution of density interface between two water layers of different salinity and the buoyancy flux produced by continuous and vertically uniform turbulent impact at a wide range of Richardson numbers, rather low Reynolds numbers, and high Prandtl number ($\approx 700$).

1. The maintaining of a density interface between turbulent layers in a sharpening mode is possible only under the condition $Ri > Ri^*$. If $Ri < Ri^*$ the density interface exists in the eroding non-stationary mode. The transition between these two modes is abrupt and the physical explanation of such a kind of transition from one mode to the other seems to be consistent with Phillips-Posmienter mechanism.

2. The maximum mixing efficiency of a stratified fluid, is achieved at $Ri \approx Ri^*$, when the thickness of the density interfaces between the quasi-homogeneous layers is in a transition between the sharpening and eroding modes. Thus, the transformation of a continuously density-stratified aqueous medium into a step-like density structure should increase the efficiency of vertical turbulent exchange.

Additional research (laboratory, numerical, field) is needed to study further this complicated problem and its relation with the step-like FS formation in the turbulent stratified fluid—oceanic pycnocline.

Finally, a few words should be said about the applications of the results of this study. The possible practical use of the results is the prediction of the evolution of the oceanic fine structure (FS), formed by the action of various physical mechanisms. One of these mechanisms is related with cross-frontal intrusions: the water layers penetrating from one side of the oceanic front to the other along the isopycnal surfaces [1,2]. If the rate of turbulence in the stratified waters surrounding the intrusion is low (the Richardson number is higher than the critical one), the intrusion can exist for a long time and spread to a large distance from the front, maintaining its boundaries in a sharpened state. In the opposite case (the Richardson number is below the critical one), the intrusion will quickly disappear (erode) due to vertical turbulent diffusion. Evidently, the cross-frontal heat and mass transfer, the regularities of which are poorly understood, should depend on the lifetime of the intrusion. The list of such examples could be continued.

**Author Contributions:** Conceptualization, A.G.Z.; methodology, A.G.Z. and V.V.G.; laboratory setup, V.V.G.; formal analysis, V.V.G. and A.G.O.; resources, A.G.Z.; data curation, V.V.G.; writing—original draft preparation, A.G.Z.; writing—editing, A.G.O.; project administration, A.G.Z.; and funding acquisition, A.G.Z. All authors have read and agreed to the published version of the manuscript.

**Funding:** The work was carried out within the framework of the state assignment No. FMWE-2021-0002 and with the support of the Russian Foundation for Basic Research grant No. 20-05-000496.

**Conflicts of Interest:** The authors declare no conflict of interest.

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
