# Peer review of "Laboratory Study of Turbulent Mass Exchange in a Stratified Fluid"

_jmse, doi:10.3390/jmse10060756_

Round 1

Reviewer 1 Report

Here are several questions about this work:

  1. What kind of situation in the real ocean can be applied for if somebody would like to use this experimental result? And what is newly findings from this study comparing to the previous study results?
  2. In Figure 1, the max S/Sav showed an increasing trend as time goes.  it is 1.2 after 1 min, however it is greater than 1.4 after 20 min around 200 mm. Since total salinity was conserved in the tank, the average salinity should be same. Then the salinity should be decreased if there was mixing happened. How was this S/Sav increased?
  3. In Figure 6a, the h1 was increased by 1000 sec, what kind of mechanism could make this increasing  layer thickness? If mixing was happened, then it is natural that the higher salinity layer should be decreased. However this plot showed opposite trend for this time period.
  4. Looking at Figure 5b, the h2 was suddenly increased from 4000 sec. But, Figure 4 showed the thickness of h2 was already increased when time is 59 min (3540 sec). Comparing these two figures, it does not match with the variation trend of thickness of h2. The authors need to make double check. 
  5. In Figure 14, the red line is moving averaged over window of 14000 values. The sampling rate of conductivity was shown about 1 Hz in line 134 (page 4). If one experimental run was carried out for 100 min, then about 6000 salinity data could be obtained. Then, how can the window of 14000 data values be used for the moving averaging? The author need to check again. -The End-

Author Response

The authors express their deep acknowledgment and gratitude to the reviewers who carefully read the paper and made many useful comments. Almost all the comments were taken into account during the revision of the manuscript and the figures.

How exactly the comments are taken into account is written in the responses to each of the reviews (see below). Also, we made a number of changes to the paper, not directly related to the reviewer comments. The main of these changes were as follows.

  1. We added to the paper a table (Table 1) which shows the values of the defining dimensional and dimensionless parameters for all 39 experimental runs performed during the laboratory study.
  2. We critically revised the definition of the turbulent velocity scale. In the previous version of the article, it was defined by formula (4) as U=4AT∙Sr/S0. In the present version, this definition is preserved. However, in the expression for Sr = 2Ad, which characterizes the turbulence generation area by oscillating rods, the factor B=2.4 is introduced as follows: Sr = 2BAd. This allowed the equation to account for increased value of U. The new value is more consistent with preliminary results of the experimental determination of the turbulent velocity scale based on the use of the Particle image velocimetry (PIV) method. We are planning to use PIV method in future for a full-fledged study of the dependence of turbulence parameters on the oscillation characteristics of the rods.
  3. The change in the turbulent velocity scale entailed a change in the values of the Reynolds and Richardson numbers. As a result, several plots were rebuilt and figures 3, 7-11 and 17 were changed. The dependencies (5)-(8), (10) and (11) were recalculated also. However these modifications did not lead to fundamental changes of the main results of our experimental study.
  1. What kind of situation in the real ocean can be applied for if somebody would like to use this experimental result? And what is newly findings from this study comparing to the previous study results?

The basic result of the experimental study is important for understanding of the conditions of formation and maintenance of the sharpened state of density interface between fine-structured layers in the presence of turbulence in a stratified sea. The new findings are as follows: a) the maximum mixing efficiency of a stratified fluid, is achieved, when the thickness of the density interface between the quasi-homogeneous layers is in transition state between the sharpening mode and the eroding mode; b) the transition between sharpening mode and eroding mode occurs according to the Phillips-Posmentier mechanism;c) the transition between these two modes is abrupt and the physical background of such a fast  transition needs further investigation.

  1. In Figure 1, the max S/Sav showed an increasing trend as time goes. It is 1.2 after 1 min, however it is greater than 1.4 after 20 min around 200 mm. Since total salinity was conserved in the tank, the average salinity should be same. Then the salinity should be decreased if there was mixing happened. How was this S/Sav increased?

Sorry, it was technical mistake and we corrected it. In the new version of the manuscript, we decided to exclude the plots with the profiles and to use only the consecutive shadowgraphs in the Figure 1 that clearly illustrate the formation of the regular step-like fine structure during long-term stirring of linearly stratified water column by horizontally oscillating vertical rods.

  1. In Figure 6a, the h1 was increased by 1000 sec, what kind of mechanism could make this increasing layer thickness? If mixing was happened, then it is natural that the higher salinity layer should be decreased. However this plot showed opposite trend for this time period.

At the beginning of each experimental run there was rather short stage during which the layered system was reaching its quasi-equilibrium state. The first three points at figure 6a indicate this initial stage. Appropriate explanation is added into the text.

  1. Looking at Figure 5b, the h2 was suddenly increased from 4000 sec. But, Figure 4 showed the thickness of h2 was already increased when time is 59 min (3540 sec). Comparing these two figures, it does not match with the variation trend of thickness of h2. The authors need to make double check.

The variability of salinity profiles at Ri close to Ri* is rather high because of strong amplitude fluctuations. This makes it difficult to determine the actual thickness of the density interface. In the improved version of the paper we replaced the profile measured at t=59 min by a previous one measured at t=55 min. It helped to illustrate the evolution of h2 layer more comprehensively.

  1. In Figure 14, the red line is moving averaged over window of 14000 values. The sampling rate of conductivity was shown about 1 Hz in line 134 (page 4). If one experimental run was carried out for 100 min, then about 6000 salinity data could be obtained. Then, how can the window of 14000 data values be used for the moving averaging? The author need to check again.

Thank you for noting this. The sampling rate of conductivity was shown about 1 Hz  for the four electrode conductivity sensor. For the single electrode conductivity sensor the sampling rate was ten times faster. So for the experimental data shown in Fig.14, the sampling rate was 10 Hz. The text is corrected to reflect this.

Reviewer 2 Report

General comments: The work is interesting and the result is applicable for further studies. It might be appropriate for publication in the JMSE Journal after performing a minor revision. The laboratory experiments in this study investigated turbulent exchange between two quasi-homogeneous layers, carried out by a system of horizontally oscillating vertical rods to generate turbulence. These results comprehensively improve the understandings of the turbulent mixing mechanisms. Therefore, I would recommend publication after minor revisions. But there are still some problems that need improvement.

Specific comments:

  • In the laboratory modeling, why the thickness of the two quasi-homogeneous layers is equal? I think it is not essential for the turbulent exchange study.
  • In Figure 1, what does the vertical axis mean? It should be explained.
  • Figure 4,5,12: Vertical axis should be reversed upside down in accordanceto other figures in the manuscript, for conveniently 
  • Line 63: Full stop should be supplemented after ‘between the layers’.
  • Line 108: Comma should be supplemented before ‘the tank’.
  • Line 250: What does the word ‘suppressing’mean?
  • Figure 7: There are wrong for the numbers of vertical axis.
  • Line 277: The word ‘reynold’should be revised to ‘Reynolds’.
  • Line 339-362: I cannot understand the corresponding, please explain in detail.
  • Line 382: Need a reference for the formula of Ri.
  • Line 402: The coefficient C is different to that in Formula (11). It should be revised to another one. And its quantity should be supplemented in Figure 17.

Author Response

The authors express their deep acknowledgment and gratitude to the reviewers who carefully read the paper and made many useful comments. Almost all the comments were taken into account during the revision of the manuscript and the figures.

How exactly the comments are taken into account is written in the responses please see below. Also, we made a number of changes to the paper, not directly related to the reviewer comments. The main of these changes were as follows.

  1. We added to the paper a table (Table 1) which shows the values of the defining dimensional and dimensionless parameters for all 39 experimental runs performed during the laboratory study.
  2. We critically revised the definition of the turbulent velocity scale. In the previous version of the article, it was defined by formula (4) as U=4AT∙Sr/S0. In the present version, this definition is preserved. However, in the expression for Sr = 2Ad, which characterizes the turbulence generation area by oscillating rods, the factor B=2.4 is introduced as follows: Sr = 2BAd. This allowed the equation to account for increased value of U. The new value is more consistent with preliminary results of the experimental determination of the turbulent velocity scale based on the use of the Particle image velocimetry (PIV) method. We are planning to use PIV method in future for a full-fledged study of the dependence of turbulence parameters on the oscillation characteristics of the rods.
  3. The change in the turbulent velocity scale entailed a change in the values of the Reynolds and Richardson numbers. As a result, several plots were rebuilt and figures 3, 7-11 and 17 were changed. The dependencies (5)-(8), (10) and (11) were recalculated also. However these modifications did not lead to fundamental changes of the main results of our experimental study.
  4. In the laboratory modeling, why the thickness of the two quasi-homogeneous layers is equal? I think it is not essential for the turbulent exchange study.

Thank you for the important question. In the manuscript, it was suggested that the parameters of turbulence in the layers do not depend on layer thickness. However if one of the layers (upper, or lower) would be sufficiently lesser than another one this proposition could be wrong. In that case the entrainment velocity from the upper to the lower layer and visa-versa could become different. Consequently the density interface should move while increasing the difference in the thickness of the quasi-homogeneous layers. So we decided to make the thickness of the both quasi-homogeneous layers equal to each other in order to hold the depth of the density interface.

  1. In Figure 1, what does the vertical axis mean? It should be explained. Vertical axis is the ratio of local salinity to average salinity of the period?

Yes, so it was. However in new version of the manuscript we decided to exclude the plots with the profiles and to use in the Figure 1 only the consecutive shadowgraphs that clearly illustrate the formation of the regular step-like fine structure during long-term stirring of linearly stratified water column by horizontally oscillating vertical rods.

  1. Figure 4,5,12: Vertical axis should be reversed upside down in accordance to other figures in the manuscript, for conveniently.

It is done.

4   Line 63: Full stop should be supplemented after ‘between the layers’.

It is done.  

  1. Line 108: Comma should be supplemented before ‘the tank’

It is done.

  1. Line 250: What does the word ‘suppressing’ mean?

We mean the word “suppressing” in sense of the reduction of turbulence at the density interface that prevents the penetration of eddies from one quasi-homogeneous layer to the other

  1. Figure 7: There are wrong for the numbers of vertical axis.

The numbers are changed.

  1. Line 277: The word ‘reynold’should be revised to ‘Reynolds’

It is done.

  1. Line 339-362: I cannot understand the corresponding, please explain in detail.

Thank you. This part of text has been revised and presented in the following form.

“The readings of the electrical conductivity sensor located in the area of the density interface, is shown at Fig. 14. Here, the red line is the smoothed data of salinity during the experimental run, and the blue line is the raw data indicating high-amplitude salinity fluctuations in a wide frequency range. Ensemble of the data shown on the graph by a double-sided arrow was taken for spectral analysis after removal of the linear trend.

The power spectrum of salinity fluctuations presented at Fig. 15 was rather smooth without pronounced peaks. This was surprising, since, in our opinion, the frequency of the rods oscillation, as well as its overtones, should have appeared. One of the possible reasons for the absence of peaks associated with the oscillation frequency of the rods may be the mutual influence of neighboring grids of rods on turbulent fluctuations, causing a masking effect.

To test this hypothesis, an experimental run was carried out where unlike the previous experimental run only one grid with rods (instead of six grids) was oscillating in a tank while the other conditions were the same. As before, an electrical conductivity microsensor was located near the oscillating grid in the area of the density interface between the layers. The spectral energy of the measured salinity fluctuations is shown at Fig. 16. Clear-cut are the peaks of energy at f=3.3*10^(-1) s^(-1)and 6.6*10^(-1) s^(-1). These frequencies corresponded well to the grid oscillation period of 3.2 s and to its double frequency overtone. This confirmed the hypothesis that the absence of peaks associated with the oscillation frequency of the rods at Fig. 14 was related with the mutual influence of neighboring grids of rods on the spectral composition of turbulent fluctuations.

As for the threshold frequency at which the spectral energy reaches its maximum values and a plateau (see Fig. 15), it is about  which corresponds to the time scale τ ≈ 30 s. If we assume that this time scale characterized the overturn period of fluid particles in energy-carrying turbulent eddies, and the particle velocity U was determined by formula (4), it was easy to obtain that Uτ ≈ pM. Thus, the spectral analysis of salinity fluctuations indirectly confirmed the validity of the velocity scale U parametrization by formula (4), as well as the fact that eddy diameter is close to M - the mesh size of the grid.”

  1. Line 382: Need a reference for the formula of Ri.

In order to separate the situation with a shear turbulent flow from a shear-less one, we introduced the shear Richardson number Ris and referred to a paper by Peligry and Sandra [15]

Line 402: The coefficient C is different to that in Formula (11). It should be revised to another one. And its quantity should be supplemented in Figure 17.

We called this coefficient C2 . It was determined that C2 =13 for asymptote at Ri>Ri* (added to the caption to Fig.17).

Round 2

Reviewer 1 Report

  1. The authors could say like that the study result could be important for any condition even though it does not exist in the real ocean. However the scope of this journal is about the ocean environment or engineering view point in the coastal or open ocean. As I think, the study in the lab experiment should have a targeting situation in the real ocean. Otherwise it is hard to say it’s worth of study. It may be worth of pure physics of fluid mechanics, so the author could find other proper journals to submit this research result.
  2. The author agreed that the figure 1 was wrong and it was mistake, and then they said the figure was corrected!. But they removed this figure in the new version of manuscript without showing the corrected figure to reviewers. I don’t understand this authors’ action (It could make reviewers doubting something wrong). At least they should show the corrected results thru response letter. 
  3. The authors did not explain the difference between two figures of figure 4 and 6. I think the authors need to define more strict definition of each layer and re-plot the figure 6, 7 and related figures and analyze the results. Glancing at the new version of figure 4, the variation thickness of layer 2 (h2) was not sudden, but smooth. 
  4. All measurement and data analysis methods need to be described in section “2. Material and Methods” including all sensors. And it is hard to find the specification of conductivity meter Expert 002  in the homepage https://magazinlab.ru/konduktometr-jekspert-002.html. In addition (minor issue), could you add when the authors carry out these experiments? -The End-

Author Response

Response to Reviewer 1 Comments.

Point 1: The authors could say like that the study result could be important for any condition even though it does not exist in the real ocean. However the scope of this journal is about the ocean environment or engineering view point in the coastal or open ocean. As I think, the study in the lab experiment should have a targeting situation in the real ocean. Otherwise it is hard to say it’s worth of study. It may be worth of pure physics of fluid mechanics, so the author could find other proper journals to submit this research result.

Response 1: The authors do not agree that the results of the study can be worth only of pure physics, or of fluid mechanics. In fact, it can be of practical importance, since it makes it possible to predict at what degree of turbulence a stratified water column could exist in the “layered” state, and at what degree it cannot remain in such state and tends to a state of continuous stratification. An example of the possible practical use of the results is the prediction of the patterns of evolution of the oceanic fine structure (FS), formed by the action of various physical mechanisms. One of these mechanisms is related with cross-frontal intrusions - the water layers penetrating from one side of the front to the other along the isopycnal surfaces (Fedorov, 1978). If the rate of turbulence in the stratified waters surrounding the intrusion is low (the Richardson number is higher than the critical one), the intrusion can exist for a long time and spread to a large distance from the front, maintaining its boundaries in a sharpened state. In the opposite case (the Richardson number is below the critical one), the intrusion will quickly disappear (erode) due to vertical turbulent diffusion. Evidently, the cross-frontal heat and mass transfer, the regularities of which are poorly understood, should depend on the lifetime of the intrusion. The list of such examples could be continued.

We have added a paragraph explaining the possible application of this research to section 5 “Conclusions”.

Point 2: The author agreed that the figure 1 was wrong and it was mistake, and then they said the figure was corrected!. But they removed this figure in the new version of manuscript without showing the corrected figure to reviewers. I don’t understand this authors’ action (It could make reviewers doubting something wrong). At least they should show the corrected results thru response letter.

Response 2: Sorry for not including the corrected Fig. 1from the first version of the paper to the response message. Now it is presented in thee atteched file.

Point 3: The authors did not explain the difference between two figures of figure 4 and 6. I think the authors need to define more strict definition of each layer and re-plot the figure 6, 7 and related figures and analyze the results. Glancing at the new version of figure 4, the variation thickness of layer 2 (h2) was not sudden, but smooth.

Response 3: Thank you for emphasizing the necessity of the sublayer’s definition. The new Figure 4 (see please in the attached file and in the updated article), shows examples of fluid column stratification at different moments of time. We have added tangents (marked in orange) to the quasi-linear sections of the profiles with maximum salinity gradient. The thickness h2 of these sections (sublayers) was determined as the vertical distance between the points of separation of the tangents from the real profiles. As a result Figure 5 shows the typical vertical salinity structure of the fluid column divided into three sublayers: two quasi-homogeneous sublayers (top and bottom) with thicknesses h1 and h3, correspondingly, and a middle sublayer (density interface) with a thickness h2 with a maximum salinity gradient. Figure 6 shows an evolution of h1 and h2 with time in details, taking in to account all the salinity profiles obtained during the experimental run. 

Point 4: All measurement and data analysis methods need to be described in section “2. Material and Methods” including all sensors. And it is hard to find the specification of conductivity meter Expert 002 in the homepage https://magazinlab.ru/konduktometr-jekspert-002.html. In addition (minor issue), could you add when the authors carry out these experiments?

Response 4: We added to the text in section 2 the following information.

Starting from line 120:  During each experimental run, the current salinity S1 in the upper layer was measured using a laboratory conductivity meter Expert 002. It is a four-electrode sensor with a diameter 0.5 cm, and a length 3.0 cm. Using six sub-ranges it provides the measurement of electrical conductivity in a range from 0.001 µS/cm up to 1000 mS/cm.The standart error of conductivity measurements do not exceed 2% (https://magazinlab.ru/konduktometr-jekspert-002.html).

Line 160: Experiment was carried out in the time period from June, 2021 to February, 2022.
